# Cryopreservation Cooling Rate Impacts Post-Thaw Sperm Motility and Survival in *Litoria booroolongensis*

**DOI:** 10.3390/ani13193014

**Published:** 2023-09-25

**Authors:** Rebecca J. Hobbs, Rose Upton, Natalie E. Calatayud, Aimee J. Silla, Jonathan Daly, Michael S. McFadden, Justine K. O’Brien

**Affiliations:** 1Taronga Institute of Science and Learning, Taronga Conservation Society Australia, Mosman, NSW 2088, Australiajobrien@zoo.nsw.gov.au (J.K.O.); 2Conservation Biology Research Group, School of Environmental and Life Sciences, The University of Newcastle, Newcastle, NSW 2308, Australia; 3Beckman Center for Conservation Research, San Diego Zoo Wildlife Alliance, 15600 San Pasqual Valley Road, Escondido, CA 92025, USA; 4School of Earth, Atmospheric and Life Sciences, University of Wollongong, Wollongong, NSW 2522, Australia; asilla@uow.edu.au; 5School of Biological, Earth and Environmental Sciences, University of New South Wales, Sydney, NSW 2052, Australia

**Keywords:** amphibian, conservation, CASA, reproductive technologies, ART, spermiation, biobanking, GRB, breeding program

## Abstract

**Simple Summary:**

The development of reproductive technologies for amphibian species has accelerated over the last decade to the point that the industry implementation of these methodologies to assist breeding and genetic management is underway globally. A conservation breeding program was established at Taronga Conservation Society (Australia) in 2019 for the critically endangered Booroolong frog (*Litoria booroolongensis*) following an emergency extraction from the wild. Shortly thereafter, a trial was undertaken to compare the efficacy of two sperm cryopreservation protocols with a view to creating a “sperm bank” from these genetically important founder individuals. Spermic urine samples from *L. booroolongensis* were successfully cryopreserved using both methods tested; however, higher numbers of sperm retained motility post-thaw when using a faster cooling method for cryopreservation. Moreover, the two cryoprotectants tested were equally effective at supporting sperm survival and function post-thaw, regardless of cooling speed. We utilized the method outlined in this paper to collect and cryopreserve spermic urine samples across three breeding seasons from all (*n* = 28) *L. booroolongensis* founder males at Taronga and are now investigating the use of these samples to create offspring via assisted fertilization.

**Abstract:**

The cryopreservation and storage of gametes (biobanking) can provide a long-term, low-cost option for the preservation of population genetic diversity and is particularly impactful when applied to manage selective breeding within conservation breeding programs (CBPs). This study aimed to develop a sperm cryopreservation protocol for the critically endangered Booroolong frog (*Litoria booroolongensis*) to capture founder genetics within the recently established (est. 2019) CBP for this species. Hormone-induced sperm release was achieved using established protocols, and spermic urine samples were collected over a 6-h period. Pooled spermic urine samples (*n* = 3 males) were divided equally between two cryoprotectant (CPA) treatments and diluted by 1:5 (sperm:CPA) with either 15% (*v*/*v*) dimethyl sulfoxide + 1% (*w*/*v*) sucrose in simplified amphibian Ringer’s (SAR; CPAA) or 10% (*v*/*v*) dimethylformamide + 10% (*w*/*v*) trehalose dihydrate in SAR (CPAB). The samples were cryopreserved in 0.25 mL straws using either a programmable freezer (FrA) or an adapted dry shipper method (FrB). The thawed samples were activated via dilution in water and assessed for viability and motility using both manual assessment and computer-assisted sperm analysis (CASA; 0 h, 0.5 h post-thaw). Upon activation, the survival and recovery of motility (total motility, forward progression and velocity) of cryopreserved sperm suspensions were higher for sperm preserved using FrB than FrA, regardless of CPA composition. This work supports our long-term goal to pioneer the integration of biobanked cryopreserved sperm with population genetic management to maximize restoration program outcomes for Australian amphibian species.

## 1. Introduction

The application of reproductive technologies in concert with the establishment of genome resource banks (biobanks) is a priority conservation management action for amphibians endorsed by the IUCN [1,2]. The strategic biobanking of gametes offers the potential for the large-scale, long-term, low-cost preservation of amphibian genetic diversity in the form of living biological material for conservation and research [3,4]. Biobanked sperm can be used for selective breeding towards a more tailored management approach and is particularly relevant to programs that bridge the gap between ex situ propagation and in situ repopulation, where in situ selective pressure will continue to drive evolution. Over the past decade, there has been an increase in the availability of assisted reproductive techniques for amphibians [5,6] and a global expansion of amphibian biobanking activities [7,8,9] in concert with the establishment of captive breeding programs (CBPs) for an increasing number of species [10,11].

In 2019, Taronga Conservation Society Australia was engaged to establish an insurance population of the genetically divergent northern populations of the Booroolong frog (*Litoria booroolongensis*) to avoid an imminent threat of extinction due to prolonged drought conditions, compounded by predictions of extreme temperatures and increasing fire danger for summer 2019 to 2020 [12,13]. *Litoria booroolongensis* is a medium-sized frog of the subfamily *Pelodryadidae* (tree frogs). Once distributed extensively across eastern New South Wales and northeastern Victoria, Australia, it is now restricted to small, isolated populations and considered critically endangered by the IUCN [14]. A collaborative and strategic collection of sixty adult frogs (*n* = 30 females, *n* = 30 males) was undertaken across the northern distribution of the species’ range to secure an ex situ breeding population at Taronga Zoo (Sydney, Australia) and establish a CBP. This species is seasonally reproductive, individuals have a relatively short lifespan of 4 years in the wild, and the males were of unknown age at collection; therefore, the rapid development of cryopreservation methodologies within the first breeding season was a high priority in order to secure founder genetics for this CBP.

In anurans, dimethyl sulfoxide (DMSO) and dimethylformamide (DMF) are both effective sperm cryoprotectants (CPAs) whose cryoprotective properties appear additive when used in combination with a non-penetrating CPA (i.e., sucrose (SUC), trehalose dihydrate (TRE); [15,16,17], reviewed in: [6]). Sperm obtained from whole testis macerates of approximately fifteen *Litoria* species has been successfully cryopreserved using 15% (*v*/*v*) DMSO with the addition of either 1 or 10% (*w*/*v*) SUC [18,19,20]. The fertility potential of sperm samples cryopreserved using this method was demonstrated via the production of sexually mature offspring in the closely related *L. fallax* [21] and *L. aurea* [20] using sperm from testicular macerates. Although high numbers of sperm are obtained from testis macerates, this lethal method is not ideal when working within CBPs for which the ongoing contribution of individuals to the population through natural breeding is desired. 

Spermiation and the production of spermic urine in amphibians can be induced via the non-lethal application of exogenous hormones [5,22], commonly a gonadotrophin-releasing hormone agonist or human chorionic gonadotrophin, which is determined through empirical testing on a species-by-species basis [22]. To date, there are no published data on the successful cryopreservation of spermic urine from *Litoria* species [23]; however, spermic urine cryopreservation and successful post-thaw recovery have been achieved in approximately 20 species globally, primarily of the Bufonidae and Ranidae families, with fertility potential of cryopreserved sperm demonstrated for eight species (see: [6]). 

In *Rana temporania*, the efficacy of the cryoprotectants DMSO and DMF in a base medium containing either 10% (*v*/*v*) TRE or 10% (*w*/*v*) SUC have been directly compared, with 12–15% (*v*/*v*) DMF supporting the superior post-thaw survival and motility of sperm collected using testis maceration [24] and hormone-induced spermiation [25]. Contrastingly, post-thaw sperm motility was statistically similar in spermic urine samples cryopreserved with either 10% (*v*/*v*) DMF + 10% (*w*/*v*) TRE or 10% (*v*/*v*) DMSO + 10% (*w*/*v*) TRE in *Anaxyrus boreas boreas*, *A. fowleri* and *Peltophryne lemur* [26,27,28], with the survival of *A. fowleri* larvae created using DMF-treated sperm higher than with DMSO-treated sperm after 7 days post-hatching [27]. Aside from clear species differences, the major methodological difference between these studies was the cooling rate employed; in *R. temporania*, a slow rate of 5–7 °C min^−1^ was used for both testicular sperm and spermic urine [24], whereas the latter studies used faster rates [26,27,28], between −20 and −45 °C min^−1^. It is common in the amphibian sperm cryopreservation literature to use a slower cooling rate (≤−10 °C min^−1^) for testicular sperm compared to spermic urine samples (≥−20 °C min^−1^) [6]. Sperm samples harvested directly from the testes are likely to have a higher percentage of immature sperm than spermic urine samples, immature sperm having not yet passed through the extra-testicular accessory structures [29], where a reduction in cytoplasmic volume and/or an alteration of membrane components is presumed to occur as in other vertebrates (e.g., [30,31]). In *Litoria* species, the cryopreservation of testicular sperm with a slower cooling rate (approximately—3 °C min^−1^) is achieved using a controlled-rate freezer [18,19].

The objective of the present study was to develop and optimize cryopreservation protocols for spermic urine samples from *L. booroolongensis*, comparing two cryoprotective agents and two cooling rates. We compared the efficacy of a previously successful methodology used in other *Litoria* species for testicular sperm using a CPA containing DMSO and SUC [18] against an alternate CPA (DMF + TRE) and slow cooling with a field-friendly faster cooling rate (approximately—20 °C min^−1^). We utilized a split ejaculate, 2 × 2 × 2 factorial design to investigate the interaction between CPA and cooling rate. From these results, we further refined the optimal protocol in an attempt to maximize sperm numbers per straw by testing whether sperm would tolerate a lower dilution ratio. We compared a 1:1 dilution in CPA to a 1:5 dilution previously used for testicular macerates in *Litoria* species [19]. Our aim was to develop a cryopreservation protocol and methodology to enable the efficient and reliable biobanking of *L. booroolongensis* spermic urine samples and to capture genetic diversity both within the CBP and from wild individuals in situ.

## 2. Materials and Methods

### 2.1. Animals

All experimental procedures were approved by the Animal Ethics Committee of Taronga Conservation Society, Sydney, Australia (approval numbers 4a1019 and 4b0820). Wild male *L. booroolongensis* (*n* = 28) of unknown age were extracted from drought-affected areas within their northern home range throughout December 2019 to January 2020 and transferred to quarantine holding at Taronga Zoo (Sydney). Males were housed individually or in same-sex pairs in plastic palpens 170 × 270 × 200 mm with filtered water, artificial plants and PVC piping for refuge. Males were fed crickets coated in calcium and multivitamin powder three times per week. Lighting was provided from 7 a.m. to 6 p.m. via ZooMed T8 10.0 UVB fluorescent tubes. The temperature within the room was maintained between 24 and 27 °C during the day and 20 and 22 °C at night. Experimental procedures were conducted during the first breeding season (September 2020–January 2021), less than 11 months after translocation from the wild.

### 2.2. Spermic Urine Collection and Evaluation 

The experiment was conducted over 1 week in November 2020. All chemicals were purchased from Sigma Chemicals (Australia) unless otherwise stated. Food was withheld 24 h prior to the collection of spermic urine to reduce the likelihood of the fecal contamination of the samples. Refined protocols for spermic urine collection in *L. booroolongensis* were applied [32,33,34]. On each experimental day, spermiation was induced in four to five males (*n* = 21 total) via the subcutaneous injection of human chorionic gonadotrophin (hCG; Choluron^®^, MSD Animal Health UK Ltd., Milton Keynes, UK) at 20 IU/g of body weight in ≤100 µL simplified amphibian ringer (3.6 mM of sodium bicarbonate, 112.5 mM of sodium chloride, 2.0 mM of potassium chloride and 1.35 mM of calcium chloride; pH 7.3–7.4; 210–220 mOsm). The males were then placed into individual plastic containers with paper towels wetted with reverse osmosis (RO) water to support hydration. Spermic urine was collected from the cloaca using fire-polished 50 µL borosilicate glass capillary tubes (Micro-caps™ Drummond Scientific Co., Broomall, PA, USA) every hour for six hours post-injection, and the samples were kept at 4 °C until assessment. Previous studies have shown spermic urine to contain maximal sperm concentration and motility between one to ten hours post-hCG-injection in this species [32]. Spermic urine samples were pooled within individuals at the three-hour time point and again at hour six (across hours four to six), and sperm quality was assessed and quantified. The pooled samples were kept at 4 °C. 

At hour three and again at hour six, the sperm motility parameters of raw spermic urine pooled within each male were assessed manually at room temperature (RT, 19–22 °C) by placing a 2 µL drop of raw spermic urine under an 18 × 18 mm coverslip balanced between two coverslips to create a vertical space and visualized at ×400 magnification with phase contrast. A minimum of 100 cells across 5 fields of view were rapidly categorized as progressively motile (forward movement regardless of direction), non-progressively motile (stationary with flagella beating) or immotile. To maximize the utilization of sample volume, membrane integrity and cell counts were performed simultaneously. Sperm cells were stained using the Live/Dead^®^ sperm viability kit (Molecular Probes Inc., Eugene, OR, USA); 2 µL of raw spermic urine were incubated with 17 µL of a 1:1000 dilution of SYBR−14 for 5 min at room temperature, and then 1 µL of propidium iodide was added and incubated for an additional one minute. The diluted samples were loaded on a Neubauer hemocytometer and immediately assessed under fluorescence (Motic BA310 Epi-LED FL microscope with a 470 nm long-pass filter; Motic Inc. Ltd., Causeway Bay, Hong Kong). A minimum of 100 cells were counted and categorized as either membrane intact (green) or damaged (red/orange). Sperm concentration was then determined from the same slide mount under brightfield at ×400 magnification with phase contrast. 

Although sperm isolated from testis macerates of *L. booroolongensis* can maintain progressive motility during cold storage for up to 21 days [35] and the spermic urine of *L. booroolongensis* for up to 6 days [34], there is a lack of studies on amphibians examining the effect of the period of time in cold storage (>0 °C) on sperm tolerance to subsequent cryopreservation. Therefore, we chose to limit the storage time to three hours post-collection to minimize potential sperm-age effects. At hour three post-injection, individual male spermic urine samples were pooled to create a multi-male sperm sample to provide sufficient volume to perform a split-sample design experiment across all treatments. Samples from three of the five males exhibiting the highest sperm concentration, motility and volume were used to create an experimental replicate. The same three males chosen at hour three were used to create a multi-male pooled sample at hour six, except for the final six-hour replicate, which required *n* = 5 males to make up the volume. The sperm quality and concentration of the inter-male pooled samples were assessed immediately prior to cryopreservation, as for the individual male samples above.

### 2.3. Sample Cryopreservation

The multi-male sperm samples pooled at hour three (*n* = 4) and hour six (*n* = 4) were each split evenly between two cryoprotectant treatments. Samples were slowly diluted by 1:5 (110 µL of sperm: 550 µL of cryoprotectant) in pre-cooled (4 °C) diluent, either 15% (*v*/*v*) DMSO and 1% (*w*/*v*) SUC in SAR (pH 7.3–7.4; ~2300 mOsm; final concentration 12.5% DMSO + 0.83% SUC; CPAA) or 12% (*v*/*v*) DMF and 12% (*w*/*v*) TRE in SAR (pH 7.3–7.4; ~1400 mOsm; final concentration 10% DMF + 10% TRE; CPAB). The treated samples were incubated for a further 10 min at 4 °C, during which time they were homogenized via gentle flicking and loaded into 0.25-mL semen straws (IMV technologies, l’Aigle, France; 100 µL sample per straw; the straws were counterweighted with 100 µL of SAR; *n* ≥ 4 straws per CPA treatment). The straws containing each CPA treatment were then divided evenly between two cryopreservation protocols (*n* ≥ 2 straws per cooling rate) using either a programmable freezing unit (Cryobath 8000, Cryologic, PL, Victoria, Australia), as per Browne [36]; FrA), or a dry shipper (DS) protocol (FrB; adapted from [26,37]). The straws were loaded into a goblet attached to a storage cane and transferred to a pre-cooled (4 °C) DS canister. The DS canister was immediately placed in the DS (Worthington Industries, Ohio, USA, Model CX100 or CXR100) with the top rim of the canister held flush with the neck of a fully charged DS for 30 s and then gently lowered to the bottom of the DS, covered and left for a minimum of 10 min. The actual cooling rates were monitored with two thermocouple probes placed inside separate 0.25-mL semen straws containing either CPAA or CPAB. All samples were transferred to liquid nitrogen and stored for at least 18 months before thawing.

### 2.4. Post-Thaw Sample Handling and Assessment 

The frozen samples were thawed and assessed over 9 days between 2022 and 2023. The straws cryopreserved using FrA were thawed by removing the straws individually from nitrogen and placing them on a laboratory bench towel at room temperature (19 to 21 °C) for 2 min. The straws cryopreserved using FrB were thawed by removing the straws from nitrogen, waving them in the air for 2 s and then submerging them directly for 5 s in a water bath containing SAR at 40 °C. All straws were wiped with a laboratory tissue to remove any residual condensation or SAR prior to the samples’ being removed to a clean Eppendorf tube.

To assess sperm quality parameters post-thaw, the sperm samples were gently homogenized via flicking the tubes, and a sub-sample was taken and diluted by 1:4 in water (10 µL of sample in 40 µL of filtered water) to lower the osmolality and stimulate motility (activation). Post-thaw motility parameters were assessed using both manual techniques and computer-assisted sperm analysis (CASA). A manual assessment of motility was conducted as described above for raw spermic urine; however, the samples were only diluted by 1:1 with (1:1000 *v*/*v*) SYBR-14 for membrane assessment to avoid over-dilution and then viewed on a plain glass slide (8 µL under an 18 × 18 mm coverslip). Simultaneously, 3 µL of activated sample was assessed by a second researcher using a Zeiss microscope (Axiolab 5, Carl Zeiss Microscopy GmbH, Oberkochen, Germany) and CASA CEROS II system (Animal Breeder, software version: v1.11.5; Hamilton Thorne Inc., Beverly, MA, USA) equipped with a 10× negative phase contrast objective (Zeiss 10× NH CEROS II 160 nm), quantifying total sperm motility, progressive sperm motility, curvilinear velocity (VCL; sum of the distance between the sperm head positions in each frame divided by elapsed time), straight-line velocity (VSL; the distance between the first and last head positions divided by elapsed time) and average path velocity (VAP; running average of the sperm head path distance divided by elapsed time) at 60 frames s^−1^ (×1.21 magnification) using standard count chamber slides (3 µL of sample and 20 µm depth, REF SC20-01-04-B, Leja, Nieuw-Vennep, The Netherlands) and settings outlined in Table 1. The sperm quality parameters of the activated samples were assessed again using both methods at 30 min post-activation. 

### 2.5. Effect of CPA Dilution Ratio on Post-Thaw Sperm Recovery 

To maximize the sperm numbers per straw, we conducted an additional experiment to test whether sperm would tolerate a lower dilution ratio, a 1:1 dilution in optimal CPA, compared to the 1:5 dilution used above.

Three males from the original cohort were utilized for the trial in the second breeding season, and spermic urine samples were collected over a total of 5 h using the methods outlined in Section 2.2 above. The samples were pooled by individual males at hour 5 and divided equally across treatments. The samples were either diluted in CPAB (12% (*v*/*v*) DMF + 12% (*w*/*v*) TRE) at the original dilution rate of 1:5 (sperm:CPA) or diluted using double-strength CPAB (dsCPAB; 20% (*v*/*v*) DMF + 20% (*v*/*v*) TRE) at a dilution rate of 1:1 to achieve the same final concentration of cryoprotectant ([CPA]_final_ 10% DMF + 10% TRE). All samples were cryopreserved using the DS method (FrB; Section 2.3). A post-thaw quality assessment was conducted, as outlined in Section 2.4.

The average post-thaw sperm parameters are presented to compare the efficacy of 1:1 versus 1:5 CPA dilution ratios; however, statistical analyses were not performed due to the low sample size (*n* = 3–5 straws).

### 2.6. Statistics

The mean concentration and motility counts for the multi-male pooled samples were compared using MS Excel data analysis and a Chi-squared test for homogeneity.

To determine the effect of the CPA treatment, cooling rate and assessment time (0 or 30 min) on post-thaw motility (manual and CASA) and the effect of CPA treatment and cooling rate on membrane integrity, we applied generalized linear mixed models (GLMM) using R (v4.2.1) (R Foundation for Statistical Computing) with the package “glmmTMB” (v1.1.7) [38] to fit betabinomial logistic regressions—interpreted as the proportion of successful cases (i.e., motile or membrane intact sperm)—with the total number of sperm counted equaling the weights for the model. We used a 2 × 2 × 2 factorial model design with the CPA treatment, cooling rate and assessment time (0 or 30 min post-thaw) as the main effects. For total motility, the three-way interaction between the assessment time, CPA treatment and cooling rate and the interaction between the assessment times and CPA and between the assessment time and cooling rate was not significant. Thus, the model was simplified via removing these interactions. For forward-progressive motility, no interaction terms were significant, and thus, all interactions were removed from the model. Due to the small volumes, the samples were only able to be assessed for membrane integrity immediately after thawing. For membrane integrity, there was a significant interaction between the cooling rate and cryoprotectant, and this interaction was included in the model. 

To determine the effect of the CPA treatment, cooling rate and assessment time (0 or 30 min) on sperm velocity, we used generalized linear models (GLM) to fit logistic regressions with a Gaussian distribution. There were no significant interactions, and thus, the models were simplified via the exclusion of these parameters. We tested 3 measures of velocity: (1) curvilinear velocity (VCL), (2) straight-line velocity (VSL) and (3) average path velocity (VAP).

Overdispersion was dealt with in all models using the sperm pool ID and number of straws as nested random effects. QQ plot residuals, distribution, dispersion and uniformity were assessed using “DHARMa” (v0.4.5) [39]. All analyses were completed using R (Version 4.2.1; [40]). Model estimated marginal means (EMMs) and 95% confidence intervals (CI) were determined for each condition and back-transformed to proportions. Odds ratios comparing treatments were also generated using the package “emmeans” (v1.4.8) [41]. All graphing was completed using the packages “ggplot2” (v3.4.3) [42] and “gridExtra” (v2.3) [43].

## 3. Results

### 3.1. Raw Spermic Urine Quality Assessment

Spermic urine samples were collected from a total of 21 male *L. booroolongensis,* with 14 males producing high-quality samples for inclusion in the cryopreservation study. 

The sperm quality metrics amongst multi-male pooled samples prior to cryopreservation were similar, except for two pools which contained significantly (*p* < 0.05) more sperm (Table 2). 

### 3.2. Effects of CPA and Cooling Rate on Sperm Quality

#### 3.2.1. Manual Assessment of Sperm Quality

Manually assessed post-thaw sperm quality metrics within each treatment did not differ significantly between assessment time 0 and 30 min following thawing and activation; thus, only time zero assessment are displayed. There was an interaction between CPA type and cooling rate on post-thaw total motility (likelihood ratio test [LRT] χ^2^(1) = 8.36, *p* = 0.004) and membrane integrity (LRT χ^2^(1) = 4.29, *p* = 0.04; Figure 1), with little difference between CPAA (DMSO + SUC) and CPAB (DMF + TRE) when the faster cooling rate (FrB) was used, and CPAB outperforming CPAA when using a controlled, slow cooling rate (FrA). For forward-progressive motility, there was a significant effect of cooling rate (LRT χ^2^(1) = 24.9, *p* < 0.001) but not of cryoprotectant or assessment time (LRT χ^2^(1) = 1.54, *p* = 0.22 and LRT χ^2^(1) = 0.68, *p* = 0.41 respectively), with the faster cooling rate (FrB) outperforming the slow cooling rate (FrA).

Odds ratios (ORs) were used to further compare the results between treatments. Under FrA cryopreservation, the use of CPAB was 1.4 times more likely to have a positive effect on total sperm motility (estimated marginal mean [EMM] = 44.3%; Figure 1A) and 1.5 times more likely to have a positive effect on membrane integrity (EMM = 47.6%; Figure 1B) than the use of CPAA (EMM = 35.9% motile; OR: 1.4, 95% CI, 1.1, 1.8; and 35.9% intact; OR: 1.5, 95% CI 1.1, 1.9, respectively). Total sperm motility and membrane integrity were not significantly different (*p* > 0.05) between CPA treatments when cooled using FrB.

There were similar proportions of total motile and forward-progressive motile sperm in the 0- and 30-min post-thaw assessments, though a significant difference was only found for total motility (Appendix A). Regardless of the cryoprotectant or cooling rate used, sperm total motility was 1.26 times more likely to be higher at 0-min post-thaw than at 30-min post-thaw (OR: 1.26, 95% CI, 1.1, 1.5). There was no difference in forward-progressive motility between the two time points (OR: 1.1, 95% CI, 0.9, 1.3). 

CPA type did not impact forward-progressive motility at either cooling rate (Figure 1C); however, FrB was 1.8 times more likely (OR: 1.8, 95% CI, 1.5, 2.2) to have a positive effect on sperm forward-progressive motility than FrA.

#### 3.2.2. Sperm Velocity

There was no significant effect of CPA type on VCL, VSL or VAP ([LRT] χ^2^(1) = 0.56, *p* = 0.45, [LRT] χ^2^(1) = 0.15, *p* = 0.70 and [LRT] χ^2^(1) = 0.39, *p* = 0.53, respectively). There was a significant effect of both cooling rate and assessment time on VCL ([LRT] χ^2^(1) = 35.22, *p* < 0.001; and [LRT] χ^2^(1) = 5.22, *p* = 0.02, respectively), VSL ([LRT] χ^2^(1) = 25.78, *p* < 0.001 and [LRT] χ^2^(1) = 6.00, *p* = 0.01, respectively) and VAP ([LRT] χ^2^(1) = 33.19, *p* < 0.001 and [LRT] χ^2^(1) = 6.48, *p* = 0.01, respectively). For all measures, FrB resulted in higher velocities and the velocities were higher at 30 min post-thaw (Figure 2). 

Curvilinear velocity (VCL) was the highest velocity measure of all three metrics assessed (Figure 2B). The difference between average VCL and VSL was larger using FrB (CPAA: 7.25 µm s^−1^ and CPAB: 7.94 µm s^−1^) than using FrA (CPAA: 2.83 µm s^−1^ and CPAB: 4.96 µm s^−1^), meaning that the sperm tracks measured post-thaw deviated less from a straight path using a fast cooling rate than using a slow cooling rate [44].

### 3.3. Cooling Rates

The average FrB cooling rate across both CPA treatments, measured from +4 to −90 °C, was −21.0 ± 2.1 °C min^−1^. The FrB curve dynamics varied between cooling “runs” (Figure 3B,D), particularly using CPAA, where supercooling was observed between −8 to −14 °C under both cooling conditions (Figure 3A,B) This phenomenon was not unique to CPAA and was occasionally observed with CPAB in subsequent experiments and during biobanking runs using FrB (Figure A1—graphs of all biobanking curves 2020–2023). 

### 3.4. Effects of Sperm Dilution Rate on Post-Thaw Sperm Quality

The sperm did not tolerate a 1:1 dilution with double-strength CPAA and did not survive thawing (DNA decondensation and cell rupture), so only the samples exposed to CPAB and dsCPAB were assessed. All sperm quality and velocity parameters tended to be higher when the sperm were diluted by 1:1 in dsCPAB (20% DMF + 20% TRE) compared to 1:5 in CPAB (12% DMF + 12% TRE; Table 3). The sperm survived cryopreservation and thawing at a higher rate when diluted by 1:1 (56.4 ± 10% intact) with dsCPAB compared to 1:5 in CPAB (22.8 ± 19% intact). Progressive and overall motility (20 ± 6% and 56 ± 8%, respectively) were higher in the samples diluted at a 1:1 ratio compared to those dilution at a 1:5 ratio (1 ± 1% and 22 ± 3%, respectively). All velocity parameters showed a trend to be higher in samples diluted by 1:1 than 1:5.

## 4. Discussion

The results presented herein demonstrate that both DMF/TRE and DMSO/SUC are suitable cryoprotective agents for the cryopreservation of spermic urine samples in *L. booroolongensis*, preserving approximately 63% of sperm viability and 70% of pre-freeze motility using the optimum protocol. Moreover, the use of a simple, field-friendly cryopreservation method (a cooling rate of approximately −21 °C min^−1^) preserved higher sperm numbers and sperm quality compared to cooling slowly in a programmable freezing machine. 

Early reports that a slower cooling rate provides optimal cryopreservation for testicular sperm in *Bufo marinus* [45] and subsequent success (≥60% post-thaw viability) when applied across a range of *Litoria* species [18,19,20] are contrary to our findings using spermic urine, where survival rates were below 50% using the same cooling protocol. A major methodological difference between those previous studies and the present study is that we prepared CPAs in SAR, rather than distilled water. Browne [46] suggests that inorganic electrolytes should be removed or their levels controlled in CPA for optimal cryopreservation. However, a variety of diluents (water, SAR, motility inhibiting substance and bovine serum albumin) have been used successfully as CPA bases across Anuran species [19,24,25,47,48]. Though sperm will deactivate once diluted in CPA, due to its high osmolality, the addition of SAR may be important in maintaining the inactive state and preserving the morphology and integrity of the mitochondrial sheath [49,50], which is essential for the maintenance of motility [51,52]. 

The use of a faster cooling rate with DMSO/SUC as CPA in the present study produced similar proportions of motile sperm (~60% total motility) post-thaw as those published previously for testis macerates in a range of *Litoria* species (reviewed in: [6]). The higher survival and motility of spermic urine samples using a faster cooling rate has been demonstrated recently in three species [53], where cooling rates in the range of −30 to −45 °C min^−1^ proved beneficial. Furthermore, higher hatching rates were obtained following assisted fertilization (AF) using sperm cryopreserved at −30 to −45 °C min^−1^ than at −20 to −29 °C min^−1^ [27], though this may be due to higher numbers of motile sperm in aliquots for AF, rather than an improvement in sperm quality or velocity, as it is unclear whether the motile concentrations of samples were matched between groups prior to AF even though a split ejaculate design was employed. Mature sperm cells collected following hormone-induced spermiation may have a greater tolerance of cryopreservation than immature sperm of the testis. Upon their exit from the testis, sperm transit through the efferent ducts to a short ductus epididymis (reviewed in [29]), and although it is not as extensive as the epididymis in other taxa (i.e., mammals), this structure likely plays a similar key role in sperm maturation (a reduction in cytoplasm and alterations of membrane constituents [30]), thus leading to differences in the tolerance of “ejaculated” sperm to the dehydration and deformation that occur during cryopreservation [54]. Alternatively, testicular sperm may require a longer equilibration period at a slower cooling rate to allow time for the optimal dehydration of the cells, as seen with *Xenopus laevis* [15]. Though species differences may exist, as studies of *R. temporania* suggest, the optimal cooling rate is similar for testicular and spermic urine [24,25], and spermic urine from *Atelopus sp*. fares better with slow cooling and thawing [55]. 

The overwhelming majority of *Litoria* sperm cryopreservation studies have used DMSO as the permeating cryoprotectant [6]; DMF has not been trialed in *Litoria* species previously and may prove a useful alternative should the optimization of AF using spermic urine be required. Studies of other Anurans directly comparing the cryoprotective properties of CPAs containing DMF or DMSO either concur with the results presented here, with no difference in post-thaw motility or fertilization rates between treatments observed [26,28], or show higher post-thaw sperm motility, fertilization and/or hatching rates using DMF [24,25].

When applying AF as a breeding strategy, it is important to optimize sperm–egg interactions. An optimal sperm concentration and sperm:egg ratio is likely influenced by many factors, including sperm morphology, the proportion of motile sperm, velocity, egg size, chemoattractants and fluid dynamics [56,57,58], all of which need to be empirically determined for each species to maximize fertilization rates. The source of sperm (testis macerates or spermic urine) and, thus, maturational state is also an important consideration for AF. In *R. temporia,* fertilization rates were higher using both fresh and cryopreserved spermic urine samples (fresh max. ~95% fertilization; cryopreserved max. 91% fertilization) than fresh testis macerates (max. ~62% fertilization) and required fewer sperm to achieve this (≥1.5 × 10^7^ mL^−1^ or min. ~8.6 × 10^4^ sperm per egg versus 2.5 × 10^7^ or min. 1.4 × 10^5^ sperm per egg, respectively; [25]). Limited data are available for *Litoria* species, as no consistent, standardized methodology for hormone-induced oviposition is available for most species [59]; however, fresh and cryopreserved sperm from testis macerates have been successfully used for AF in both *L. aurea* (fresh ~86% fertilization, 2.75 × 10^6^ sperm ml^−1^, ≥2.75 × 10^3^ sperm per egg; cryo ~66% fertilization, 4.2 × 10^5^ to 2.5 × 10^6^, ≥4.2 × 10^2^ sperm per egg; [21]) and *L. fallax* (fresh ~86% fertilization; cryo ~36% fertilization; both at 1.2 × 10^4^ sperm per egg; [20]).

Maximizing the fertilization potential of cryopreserved sperm samples could be improved by increasing the number of motile sperm per egg for AF, as demonstrated in *A. fowleri* [60] and *B. marinus* [61]; though achieving this in *L. booroolongensis* using cryopreserved spermic urine may be challenging with initial concentrations of raw spermic urine as low as 3 × 10^6^ sperm mL^−1^ per male [32], which is further diluted during cryopreservation and post-thaw activation, before AF. Not only will a lower dilution rate with CPA improve overall sperm numbers in cryopreserved samples, but our preliminary experiments have also shown that reducing the dilution rate of spermic urine in CPA, from 1:5 to 1:1, tended to improve the proportion of motile sperm and sperm velocity post-thaw. Interestingly, this does not appear to be the case for fresh testicular sperm samples in this species, which retain motility and velocity between 1 and 6 h of collection when diluted at a ratio of 1:16 [32]. Like our post-thaw dilution findings, the dilution of fresh spermic urine samples from *Atelopus zeteki* [62] and *Anaxyrus fowleri* [60] showed significantly decreased motility within fewer than 60 min post-dilution at 1:1 and 1:5, respectively, compared to undiluted samples; though this is likely due to the rapid expenditure of energy reserves following hypo-osmotically induced activation [56], rather than dilution itself. Moreover, there may be a subtle difference in motility “type” contributing to fertilization success, as seen in *B. marinus*, for which diluted samples containing similar total sperm motility had differential fertilization success [61].

Sperm are cryopreserved in high-osmolality CPAs and require activation prior to AF, as performed in the cryo-AF studies mentioned above. This is achieved via the dilution of samples with pure water to lower the osmolality [56,63], though complete activation may involve other egg-related factors, as seen in *Cynops pyrrhogaster* [64,65], *Pelophylax shqipericus* [66], *Xenopus laevis* [67,68] and *Bufo b. japonius* [69]. Increasingly, investigators are utilizing computer-assisted sperm analysis (CASA) for sperm quality assessment in Anurans [70], which may provide further detail on the fertility potential of raw and cryopreserved samples. The cooling rate significantly affected sperm velocity in response to activation post-thaw in *L. booroolongensis* spermic urine samples, with the average VCL in the two most effective cryopreservation treatments at approximately 10 µm s^−1^. Interestingly, the VCL measured during the dilution rate trial was higher (~21 µm s^−1^) in samples diluted by half with CPA and closer to values previously recorded for raw spermic urine in this species (20–30 µm s^−1^; [32,34]). Unfortunately, we did not have the capacity to collect raw sperm data using CASA until 2022; exploring the impacts of cryopreservation and dilution on sperm velocity will be interesting to explore further.

## 5. Conclusions

Our study shows that DMF/TRE is as effective as the DMSO/SUC cryodiluent previously used to preserve testicular sperm in *Litoria* species, but spermic urine collected via hormone induction in *L. booroolongensis* requires faster cooling rates than those previously reported for testicular sperm. Since undertaking this study, we have reduced the dilution rate of sperm in cryoprotectant to increase sperm concentration in preserved samples. This, coincidentally, had a positive impact on sperm motility and velocity, which will prove invaluable for biobanking and later sample use. Biobanking requires balancing efficient storage (space requirements) with protocols that secure optimal post-thaw sample quality. Further refinements to the current cryopreservation protocol (i.e., improvements to cooling and thawing rates to further increase survival and motility post-thaw) will be made following the optimization of AF parameters using cryopreserved sperm (i.e., sperm concentrations for cryo-units; sperm–egg ratio).

This work supports the long-term goal to establish a functional amphibian biorepository within Australia’s leading wildlife biorepository, Taronga’s CryoDiversity Bank, and to pioneer the integration of sample use with population genetic management to maximize restoration program outcomes for Australian species. The relatedness of founder individuals was established in early 2020 using genome-wise single-nucleotide polymorphism (SNP) data analysis. In order to capture the maximum genetic diversity of the population of founders from this short-lived species (3–4 years), sperm collection attempts were initiated in spring 2020. Since that time, we have successfully collected and cryopreserved samples (*n* ≥ 3 samples) from all twenty-eight founder males within the population; particularly valuable are samples collected from two males who have since reached their natural lifespans and did not breed whilst at Taronga (2020–2022).

## Figures and Tables

**Figure 1 animals-13-03014-f001:**
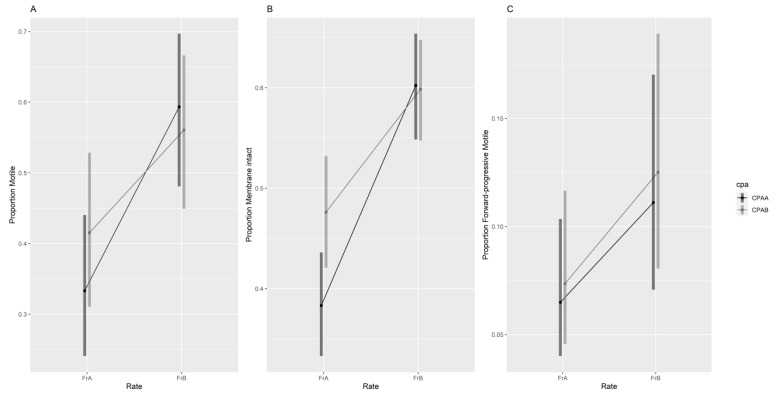
Effects of cryoprotectant and cooling rate on sperm total motility (**A**), membrane integrity (**B**) and forward-progressive motility (**C**) at 0 min post-thaw (manual assessment). Sperm motility was assessed in hypo-osmotically activated samples. FrA—programmable, FrB—dry shipper, CPAA—DMSO/SUC, CPAB—DMF/TRE.

**Figure 2 animals-13-03014-f002:**
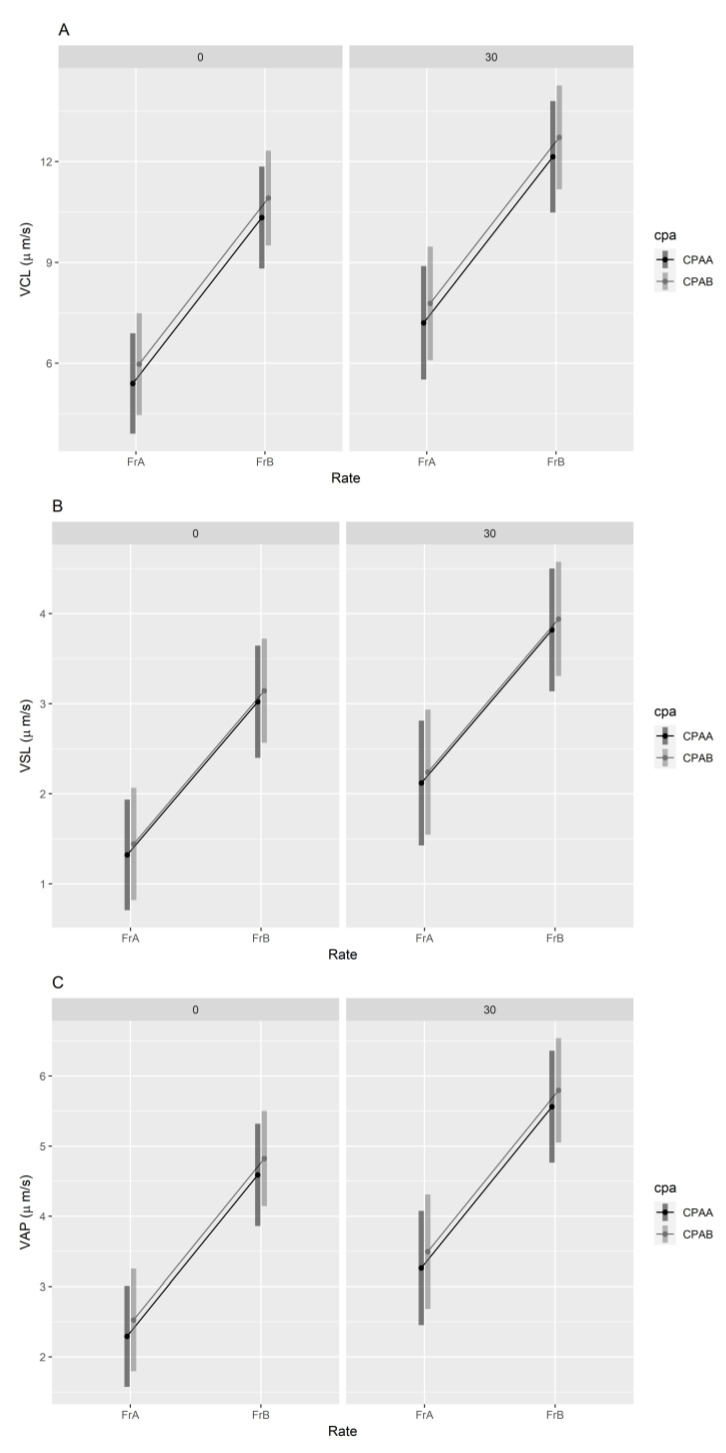
Effects of cryoprotectant and cooling rate on sperm velocity parameters at times 0 and 30 min post-thaw (CASA assessment). (**A**) VCL, curvilinear velocity; (**B**) VSL, straight-line velocity; (**C**) VAP, average path velocity. FrA—programmable, FrB—dry shipper, CPAA—DMSO/SUC, CPAB—DMF/TRE.

**Figure 3 animals-13-03014-f003:**
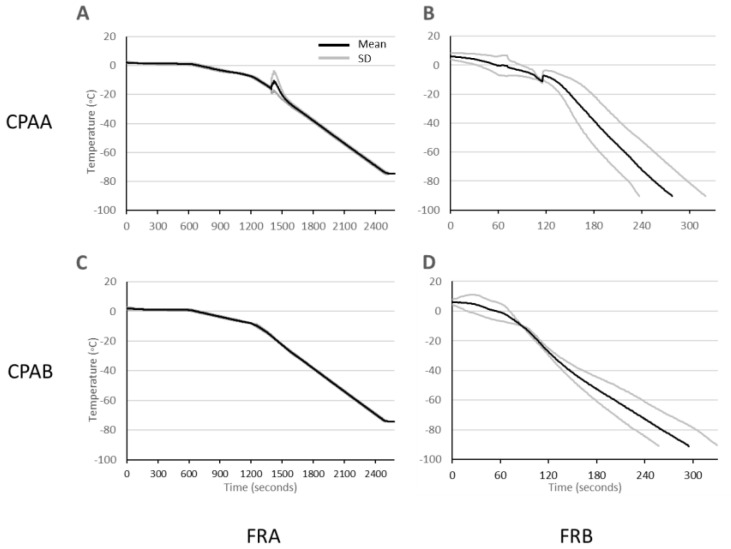
Average measured cooling curves (mean ± SD) across all experimental replicates. Straws containing CPAA (**A**,**B**) show a defined point of thermal fusion between −8 and −14 °C under both cooling conditions (FRA (**A**), FRB (**B**) compared to CPAB (**C**,**D**).

**Table 1 animals-13-03014-t001:** Hamilton Thorne CEROS II CASA settings for assessment of *L. booroolongensis* spermic urine samples.

Parameter	Set Point
Camera	
Exposure (ms)	4
Gain	300
Integrate Enabled	False
Integrate Time (ms)	500
Cell Detection	
Elongation Maximum (%)	30
Elongation Minimum (%)	1
Enable Advanced Tail Detection	False
Head Brightness Minimum	150
Head Size Maximum (µm^2^)	45
Head Size Minimum (µm^2^)	5
Static Tail Filter	False
Tail Brightness Minimum	50
Tail Min Brightness auto Offset	10
Tail Min Brightness Mode	Auto—First Frame
Chamber	
Capillary Correction	1.3
Illumination	
Histogram Smooth Width	0
Max Photometer	70
Min Photometer	50
Kinematics	
Cell Travel Max (µm)	10
Enable Motile Static Collision Avoidance	False
Motile Cells Require a Tail	False
Motile Require Tails Max VSL (µms^−1^)	0
Progressive STR (%)	0
Progressive VAP (µms^−1^)	5
Slow VAP (µms^−1^)	2
Slow VSL (µms^−1^)	0
Static Algorithm	Width_Multiplier
Static VAP (µms^−1^)	0
Static VSL (µms^−1^)	0
Static Width Multiplier	0.3

**Table 2 animals-13-03014-t002:** Sperm assessment metrics (prior to cryopreservation) for multi-male pooled spermic urine samples.

Male Group	A	A	B	B	C	C	D	DE *
Number of males in pool (*n*)	3	3	3	3	3	3	3	5
Collection pool (hours post-injection)	1–3	4–6	1–3	4–6	1–3	4–6	1–3	4–6
Sperm concentration (×10^6^/mL)	15.2	10.1	21.0	17.9	14.5	10.5	38.5 ^a^	63.0 ^b^
Progressive motility (%)	82%	72%	85%	80%	41%	85%	75%	73%
Total motility (%)	87%	78%	93%	84%	85%	92%	92%	91%
Membrane intact sperm (%)	92%	NA	93%	NA	97%	97%	89%	91%

NA, not assessed due to low volume; * an additional two males “E” were added to this pool to increase sperm volume for the replicate; superscript letters indicate significant differences within metric.

**Table 3 animals-13-03014-t003:** Post-thaw sperm quality parameters in samples diluted at different ratios with CPA prior to cryopreservation.

	Pre-Freeze	Post-Thaw	Post-Thaw
[CPA] _initial_	-	20% DMF + 20% TRE	12% DMF + 12% TRE
Dilution ratio	0	1:1	1:5
Replicates (*n* straws)	3	5	4
Total motility (%)	90.7 ± 6.7%	56.2 ± 8.3%	21.7 ± 2.8%
Progressive motility (%)	68.3 ± 19.4%	19.6 ± 5.7%	1.0 ± 0.8%
Intact cells (%)	86.3 ± 3.5%	56.4 ± 10.0%	22.8 ± 19.4%
VAP (µms^−1^)	-	10.4 ± 5.4	4.5 ± 3.3
VCL (µms^−1^)	-	19.7 ± 8.9	11.7± 11.4
VSL (µms^−1^)	-	7.7 ± 4.6	3.6 ± 2.2

## Data Availability

The data presented in this study are available on request from the corresponding author. The data are not publicly available, in accordance with Taronga Conservation Society Australia’s policies on data and sample sharing (Opportunistic Sample Request Policy).

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
