# Peer review of "Cryopreservation Cooling Rate Impacts Post-Thaw Sperm Motility and Survival in Litoria booroolongensis"

_animals, 2023, doi:10.3390/ani13193014_

Round 1

Reviewer 1 Report

Please see the comments attached below.

Author Response

We thank reviewer 1 for their time in reviewing our submission.

In response to comments that the dilution experiment seems out of place, we contend that this was an important methodological refinement of the larger study. Though not statistically compared, we showed that pre-dilution of samples prior to cryopreservation negatively impacts post-thaw quality. It is likely that "biobankers" may desire to dilute their samples to a standardised concentration prior to cryopreservation and should be aware that this can negatively impact post-thaw sample quality and should be determined for each species.

We have altered sections of text to represent the study as a methodological refinement, rather than pilot. Since completion of the initial trials, we have incorporated the lower dilution rate into our biobanking standard operating procedure.

Reviewer 2 Report

This research paper evidences the comparative success of a simple technique to cryopreserve amphibian sperm in conservation breeding programs or in the field, illustrated by its use to perpetuate the genetic diversity of a Critically Endangered species. The paper is very well designed and conducted except for a few minor issues. In particular, the use of computer aided sperm assessment is important as this technique should be generic in elaborate studies of sperm cryopreservation. However, the paper lacks a strong generic statement of its significance. The authors example is the most comprehensive for any amphibian, reptile, mammal, or bird, and is especially significant because of these taxon amphibians are subject to the most dynamic programs for the development and application of sperm banking for perpetuating genetic diversity.  I have made suggestions on the attached PDF, about how to give their achievement the merit it deserves within the bigger concept of gene banking. I have offered some suggestions for improved use, or proper use of terminologies, that would give the article a more accurate and contemporary presentation. There is the short presentation of a preliminary study that perhaps would be best left out.

The paper is fine except for some minor grammatical changes.

Author Response

We thank reviewer 2 for their constructive comments and time taken to review our submission.

We have made the majority of suggested grammatical and smaller changes in text. The more detailed comments are addressed below.   

Ln 115 - this should be in discussion, as shown in this study

Response - We have removed this sentence as suggested.

Ln 140 - What does this add? Surely all manipulations were under quarantine procedures as assumed for CBPs and handling of wild amphibians. Please clarify or remove.

Response - Whilst it may not be necessary to include that studies were conducted in quarantine (deleted text), the authors believe it is important to highlight the period of time frogs spent acclimatising to ex situ conditions and demonstrate that intervention at the first breeding season following capture did not negatively impact the animals, sperm quality, or natural fertility. These animals have all produced viable young through natural breeding whilst at Taronga.

Ln 167 - the percentage of swimming, activated and immobile, and inactivated sperm, [these terms seem more appropriate]. The term progressive motility could have application where the sperm were shown to be progressing somewhere, such as targeting an oocyte, but even in this case progressive swimming would seem better.

Response - We disagree with the proposal that progressive motility is ambiguous or incorrect and refers only to directional movement due to chemoattraction. The categories we have chosen to assign and the terms used to describe them (progressive, non-progressive, total and immotile) are standard nomenclature amongst reproductive literature to quantify sperm motility; more specifically, these categories are outlined by the World Health Organisation as standard metrics for the assessment of human fertility (World Health Organization. WHO Laboratory Manual for the Examination and Processing of Human Semen, 6th ed.;WHO Press: Geneva, Switzerland, 2021. Available online: https://www.who.int/publications/i/item/9789240030787.

Moreover, we have compared the use of manual assessment to computer-assisted sperm analysis (CASA) assessment; these categories being standard for CASA software to delineate subpopulations of sperm within a sample and further defines the types of progressive motility through analysis of velocity parameters which include analysis of swimming speed and direction.

Also, the absence of motility does not necessarily indicate inactive sperm, but also includes damaged and dead cells. This is likely the case in this study given that PI staining (indicative of damaged membranes) closely approximated immotile sperm numbers.

Category descriptors have not been changed.

Ln 203 - with straws ....pre-loaded with 100 μl SAR to [do what]????, thus providing ....≥ 4 straws ....

Response - It is standard practice to counter-weight samples in straws so they don't float in liquid nitrogen. Text has been altered for clarity.

 Ln 253 - NOTE: 1:5 of original cryoprotectant gives a final cryoprotectant dilution rate of 5/6 of 15% or 12%; and 1:1 of double cryoprotectant gives a final cryoprotectant dilution rate of 1/2 of 30% or 15%, so they are slightly different. But no big deal as long as this is mentioned here and in discussion. Later I mention that this study might be better left out.

Response - All 3 reviewers had comments regarding the pilot study. We contend that this is an important methodological refinement of the larger study. Though not statistically compared, we have shown that pre-dilution of samples in diluent prior to cryopreservation impacts post-thaw quality. It is likely that "biobankers" may desire to dilute their samples to a standardised concentration prior to cryopreservation and should be aware that this can negatively impact post-thaw sample quality and should be determined for each species.

 Further to concerns about the CPA final concentration being different for the two treatments 1:1 and 1:5. We adjusted the initial concentration of each CPA so that the final concentration would be equal across both dilution treatments. Our initial test run with CPAA caused sperm DNA decondensation, so we didn't pursue this treatment group further. We have refined the method section 2.5 to remove any confusion and have adjusted subsequent discussion sections accordingly.

We have edited the sections relating to the pilot study for clarity and included it as a refinement, rather than “pilot”. See text insertion at line 125  “From these results, we further refined the optimal protocol in an attempt to maximize sperm numbers per straw by testing whether sperm would tolerate a lower dilution ratio. We compared a 1:1 dilution in CPA to a 1:5 dilution previously used for testicular macerates in Litoria species  ”

Ln 365 - This comparative effect is quite dramatic especially when considering that 1:1 dilution is the convention in many successful studies and begs the question of why 1:5 was used in the first place, the conc is so close to the 12% and the results are so different, and why is CPAA so different. Because of the complexity of these questions this study could simply be left out.  In terms of increased concentration the authors could present this in the discussion.

Response - See comment above.

Ln394 - The term mitochondrial vesicle is a misnomer both because of scientific precedence in naming, and the fact there is no vesicular structure on the sperm head except for the acrosome.

suggested text 'including the integrity of the mitochondrial sheath that is essential for motility [Lee and Jamieson, 1993]. Lee, M.S.Y.; Jamieson, B.G.M. The Ultrastructure of the Spermatozoa of Bufonid and Hylid Frogs (Anure, Amphibia): Implications for Phylogeny and Fertilisation Biology. Zool. Scripta. 1993, 22, 309-323.

Response - Our understanding of the current literature is that the mitochondrial sheath describes the membrane over the midpiece region containing the mitochondrion. In the contemporary papers cited here and earlier descriptions of the specialised "cytoplasmic droplet” or "corpuscle", the term "mitochondrial vesicle" describes an intact mitochondrial sheath with swelling; a distinct feature of the mid-piece in many amphibia and distinct from potentially non-functional cytoplasmic droplets in many other taxa.

To remain impartial, and until the upcoming paper reference provided by reviewer 2 is published in the literature we have changed the term to "specialized cytoplasmic droplet" and the alternate names of sheath or vesicle to aid the reader in searching for additional literature of interest.

 Ln 401 - In vitro fertilisation is the conventional term. Also, artificial fertilisation (AF) is sometimes used generically for in vitro fertilisation, sometimes even disregarding including artificial insemination. It would be best to simply avoid the acronym and use in vitro throughout.

Response - Opinions are divided amongst the amphibian reproductive science community on the use of the term "assisted" versus "in vitro" fertilization to describe this technique. The authors have consistently used the term "assisted" fertilisation for studies we have published in amphibians and believe this is accurate when describing this technique for an externally fertilisation species. To remain consistent with our previous publications, including in the current special issue, we prefer to keep the nomenclature "assisted".

Ln 428 – Longevity is very important and not addressed in this study. Makes discussion of concentrations between fresh, spermic urine, and testicular macerates difficult within the current literature.

Response -  Although we didn't test longevity directly in this study, literature are available for longevity of motility in the study species (Silla 2017, Keogh 2017) and we did assess sperm motility at both 0 and 30 minutes post-thaw as we believe this will be the window of time it will take for processing of cryopreserved sperm for AF.

Albeit, taking into consideration the comments of reviewer 2 regarding the under-reporting of timing of application of sperm, we have exchanged longevity with proportion motile sperm which will be lower, more variable and likely of more importance to success of cryo-AF.

Ln 429 - NOTE: These studies can only be compared if the application time of the sperm is factored in. For instance, with L. tasmaniensis {authors spelling incorrect] the application time at a known concentration of free swimming sperm was 30 min, but the application time and type was shorter and the free swimming concentration was undetermined for other species. A review in progress shows that fertilisation curves over time give about the same fertilsation rates for the for the reviewed species irrespective of sperm concentration, and even 4 x 106 swimming fresh L. booroolongensis sperm should saturate fertilisation. However, methods vary a lot so more research is needed. Best left out. Also using L. for Limnodynastes is ambiguous with the use of L. for litoria so check for this problem throughout.

Response - These data and citations were included as examples only, to show the range of concentrations that have been tested across taxa. We understand Reviewer 2’s reservations about making any direct comparisons between these studies and have removed this sentence, as suggested.

Ln 483 - Maybe remove this because of the problems introduced earlier. However, the concept that a lower dilution rate with higher concentration of cryoprotectant could result in the same final cryoprotectant concentration and increase sperm concentration to achieve hight fertilisation rates.

Response - See comments above

Ln 495 - Congratulation! Here you have the most comprehensive sperm Banking program globally for the perpetuation of genetic diversity in any amphibian that I am aware of. Generic formulas, depend on the number of females, the amount of variation you want to protect, the type of variation (heterozygosity or alleles or quantitative variation in traits) you aim to protect, the likelihood of successful cryopreservation and subsequent successful utilization of the frozen gametes, the clutch size, and more considerations. The basic number is 20 individuals for 97.5% of genetic diversity. As a starting point, the proportional heterozygosity (or “gene diversity”, G) obtained with a sample of n initial breeders is:

Response - We thank reviewer 2 for their generous comments regarding the world-leading progress we have made in the application of these technical advances to our conservation program. We also thank Reviewer 2 for providing reference points to discuss the genetic management implications of the sampling strategy we have employed. As we have not yet tested the fertility of these cryopreserved samples (we will be undertaking experiments in coming weeks) we are reluctant to overstate the significance of this collection and prefer to leave the conclusions as written.

We will, however, consider adding these points to future publications we have planned for fertilisation experiments and their impact on further refinement to our cryopreservation protocol (i.e. packaging and minimum sample quality parameters).

Reviewer 3 Report

Hobbs et al. present here the results of their study on sperm motility and survival parameters of Litoria booroolongensis after different cryopreservation protocols. The study is overall very well described and the conclusions are supported by the results.

While the main study is generally very well described, I am having difficulties to understand the results of the pilot study (3.3). It is mentioned that the sperm did not survive double concentrated CPA, does this mean the rest of the study was performed in normal concentrated CPA just diluted 1:1 instead of 1:5? This would be in contradiction to the corresponding method section, where it is said that this was done under double CPA concentration.

In general, it would improve readability if final concentrations of CPAs would be stated in the results/figures throughout the manuscript.

Does table 3 show results for CPAA? If so, why is CPAB not shown in the table, but discussed in the text?

As a minor point, some of the numbers and labels in the figures are hard to read and the grey scale coding is difficult to distinguish in some cases.

Author Response

We thank reviewer 3 for their time in reviewing our submission.

We have refined section 3.3 to expand and clarify the methodology. To achieve the same final concentration of CPA across treatments in the dilution experiment we altered the initial concentration of the CPA correspondingly; this detail was missing, and we see how this made interpretation confusing. We only used CPAB for the dilution experiment and thus have only displayed data related to CPAB as our initial attempts to use concentrated CPAA destroyed the sperm.

These details have been added to section 3.3 and table 3.

The final concentration of CPA was consistent across the cooling rate experiment and the final concentration of CPAs are defined in the methods section. Aside from Table 3, we do not believe it necessary to add the final concentration of CPAs to the remaining figures.

We have reconfigured Figure 1 for readability.

Round 2

Reviewer 2 Report

The 'mitochondrial vesicle' does not exist and publishing this term is scientific misinformation. In respect to L405-406 and the term "specialised mitochondrial droplet" and reference to the 'mitochondrial vesicle" in brackets, use the proper scientific term as used in Lee and Jamieson, 1993, for species in the same genus as the articles target species. The cytoplasmic droplet is a residual structure containing cytoplasm and sometimes even a nucleus, from sperm formation found on immature sperm as shown as the 'accessory cell' in Waggener and Carrol, 1998. The concept of 'mitochondrial vesicle' was elaborated by Kouba et al. 2003, in a poorly researched and referenced article, with results and discussion that do not correspond with references to Waggener and Carroll 1998, in terms of structure or motility. Kouba et al. 2003, and Waggener and Carroll, 1998, both show that sperm can be motile with out the fictitious 'mitochondrial vesicle', as later presented in the literature as essential for motility.

Author Response

We have readdressed comments regarding the anatomical terminology and changed to "mitochondrial sheath", as suggested by the reviewer. We have chosen to keep the additional references (same paragraph) that describe the mitochondrial sheath's role in the maintenance of motility although they use a different term to describe it.  

The remaining format queries will be referred to the type setter for confirmation.